# Nursing Interventions for Patient Empowerment during Intensive Care Unit Discharge: A Systematic Review

**DOI:** 10.3390/ijerph182111049

**Published:** 2021-10-21

**Authors:** Cecilia Cuzco, Rodrigo Torres-Castro, Yolanda Torralba, Isabel Manzanares, Pilar Muñoz-Rey, Marta Romero-García, Ma. Antonia Martínez-Momblan, Gemma Martínez-Estalella, Pilar Delgado-Hito, Pedro Castro

**Affiliations:** 1Department of Medical lntensive Care Unit, Hospital Clinic of Barcelona, Institut d’Investigacions Biomèdiques August Pi i Sunyer (IDIBAPS), University of Barcelona, 08036 Barcelona, Spain; ccuzco@clinic.cat (C.C.); pcastro@clinic.cat (P.C.); 2Department of Fundamental Care and Medical-Surgical Nursing, Nursing School of Faculty of Medicine and Health Sciences, University of Barcelona, 08036 Barcelona, Spain; 3Department of Pulmonary Medicine, Hospital Clínic–Institut d’Investigacions Biomèdiques August Pi i Sunyer (IDIBAPS), University of Barcelona, 08036 Barcelona, Spain; rhtorres@clinic.cat (R.T.-C.); yotorra@clinic.cat (Y.T.); 4Department of Physical Therapy, Faculty of Medicine, University of Chile, Santiago 8380453, Chile; 5Biomedical Research Networking Centre on Respiratory Diseases (CIBERES), 28029 Madrid, Spain; 6Department of Epilepsy, Hospital Clinic of Barcelona, Institut d’Investigacions Biomèdiques August Pi i Sunyer (IDIBAPS), University of Barcelona, 08036 Barcelona, Spain; imanzana@clinic.cat; 7Epilepsy Study Group of the Catalan Neurological Society, Spanish Epilepsy Society, European Reference Network (ERN), 08036 Barcelona, Spain; 8Hospital Universitari Germans Trias i Pujol, 08916 Badalona, Spain; mpmunozr.germanstrias@gencat.cat; 9GRIN-IDIBELL—Nursing Research Group, Department of Fundamental Care and Medical-Surgical Nursing, Nursing School of Faculty of Medicine and Health Sciences, University of Barcelona, 08907 Barcelona, Spain; martaromero@ub.edu (M.R.-G.); mmartinezmo46@ub.edu (M.A.M.-M.); 10GRIN-IDIBELL—Nursing Research Group, Nursing Direction, Hospital Clinic of Barcelona, 08036 Barcelona, Spain; gemma@clinic.cat

**Keywords:** patient empowerment, patient education, patient information, intensive care unit discharge, intensive care unit transition, nursing interventions, systematic review

## Abstract

Intensive care unit discharge is an important transition that impacts a patient’s wellbeing. Nurses can play an essential role in this scenario, potentiating patient empowerment. A systematic review was conducted using the Preferred Reporting Items for Systematic Reviews and Meta-Analyses (the PRISMA Statement. Embase), PubMed/MEDLINE, CINAHL, Cochrane Central Register of Controlled Trials (CENTRAL), CUIDEN Plus, and LILACS databases; these were evaluated in May 2021. Two independent reviewers analyzed the studies, extracted the data, and assessed the quality of evidence. Quality of the studies included was assessed using the Cochrane risk-of-bias tool. Of the 274 articles initially identified, eight randomized controlled trials that reported on nursing interventions had mainly focused on patients’ ICU discharge preparation through information and education. The creation of ICU nurse-led teams and nurses’ involvement in critical care multidisciplinary teams also aimed to support patients during ICU discharge. This systematic review provides an update on the clinical practice aimed at improving the patient experience during ICU discharge. The main nursing interventions were based on information and education, as well as the development of new nursing roles. Understanding transitional needs and patient empowerment are key to making the transition easier.

## 1. Introduction

The number of critically ill patients has increased during the last decades. In Spain, 240,000 adults are admitted into intensive care units (ICU) each year [1]. Patients with potential life-threatening processes and vital organ dysfunction who require specialized and continuous care are admitted to the ICU [2], of whom >90% survive ICU admission [3]. Recently, ICUs have become essential in caring for seriously-ill patients admitted due to the COVID-19 pandemic [4]. The ICU plays an important role in the care process of many patients. Once patients are sufficiently stable and care can be stepped down, they can be discharged to the general ward, providing continuity of care.

Discharge or transition of the patient from the ICU to a general ward is one of the most challenging, high-risk, and inefficient care transitions because patients who are among the most seriously ill are transferred from high-tech units to less acute environments, which involves many professionals in the exchange of information and responsibilities [5]. ICU discharge is therefore a complex process, and patients’ feelings and perspectives, including a sense of displacement, anxiety, and loss of autonomy, are crucial factors [6,7]. Patients feel powerless in this context, and the lack of medical knowledge and loss of control over one’s body are seen as the main factors behind these thoughts [8].

Furthermore, these patients’ feelings and perceptions during ICU discharge could increase the risk of post-ICU syndrome (PICS). PICS can develop due to mental and cognitive impairments, physical disabilities, and psychological factors (anxiety, depression, and post-traumatic stress disorder (PTSD)) [9]. Effective interventions in ICU survivors are essential to decrease negative outcomes and increase the quality of life. Needham et al. (2012) suggested that effective interventions in patients to improve long-term outcomes after ICU discharge should focus on early psychological intervention, early mobility programs, post-discharge follow-up programs, ICU diaries, healing care environments, functional reconciliation, and the ABCDEFGH bundle (Airway management, Breathing trials, Coordination of care and Communication, Delirium assessment, Early mobility bundle, Family involvement, Follow-up referrals and Functional reconciliation, Good handoff communication, and Handout materials on PICS and PICS in Family (PICS-F)) [10,11] which addresses the risks factors for PICS, sedation, delirium, and immobility. Therefore, preparation of the ICU discharge process to carry it out accurately and correctly could be the cornerstone of a decreased risk of PICS afterwards.

In this sense, the nurses within the multidisciplinary team of the ICU develop a fundamental role in the ICU transition planning process, as they are the ones who participate in, organize, and carry out the direct interventions of patient care during the transition [12]. Thus, it is the nurses’ responsibility to assess the needs of patients during the transition and provide adequate information and education to the patient and family. To improve the efficiency of the role of ICU nurses in patients during the transition of ICU patients, some hospitals have even introduced a new nursing role called “liaison nurse” [13,14]. The competent role of ICU nurses in planning and directing the implementation of a multidisciplinary program during ICU transition that could reduce ICU readmission and hospital mortality has also been highlighted [15].

Another way in which nurses can begin to recover power to their patients is to be aware of signs and symptoms that indicate feelings of powerlessness [16]. Empowerment is a complex, multi-dimensional concept [17,18,19,20,21,22] that was introduced to allow patients to shed their passive role and play an active part in the decision-making process of their health and quality of life [20]. Successful empowerment occurs when patients come to terms with their threatened sense of security and identity, and they have a sense of control over their lives [8,23]. The benefits of improving empowerment are extensive, including decreased levels of distress and strain, an increased sense of coherence and control over the situation, and personal development and growth, together with increased comfort and inner satisfaction [24].

Patient empowerment could be a useful tool to reduce the stress associated with ICU discharge. Empowerment strategies have increased over recent years, mainly self-care in chronic illness such as diabetes [25], cancer, [26,27] and other clinical scenarios [28,29]. However, their role in ICU discharge is less well known. Although this is the responsibility of the multidisciplinary healthcare team, this transition, including patient empowerment, is usually carried out by nurses [30,31].

We conducted a systematic review to provide evidence on patient empowerment interventions, identify remaining gaps, and suggest directions for future research and clinical practice during ICU discharge. The main aim was to determine the effects of nursing interventions to improve patient empowerment during ICU discharge in adult patients.

## 2. Materials and Methods

### 2.1. Design, Protocol, and Registration

We performed a systematic review using the Preferred Reporting Items for Systematic Reviews (PRISMA) guidelines [32] (Appendix A). The review was registered in the International Prospective Register of Systematic Reviews (PROSPERO) CRD42021254377 (Appendix A).

### 2.2. Criteria for Inclusion and Exclusion in the Review

We included randomized controlled trials (RCTs) in adults of both sexes undergoing ICU discharge. The studies included aimed to determine the effect of nursing empowerment interventions on patient wellbeing. Studies were included according to the population, intervention, comparison, and outcome (PICO) criteria (P: adults during ICU discharge; I: patient’s empowerment interventions performed by nurses; C: no intervention; O: physical and mental health symptom, patient satisfaction, and readmission). Accordingly, all studies had at least one group of patients with a nursing empowerment intervention and another with usual care during ICU discharge. The nursing empowerment intervention was defined as information, behavioral instructions, and advice on the management of ICU discharge by verbal, written, audio, or video-taped means. Studies in groups of adults at ICU discharge were also included.

We included studies if they fulfilled the following criteria, (1) original research; (2) patients’ admission to the ICU; (3) reported impact of the nursing intervention; (4) full text available, without language restriction. We excluded studies with patients under 18 years of age. In addition, all observational studies, editorials, letters to the editor, review articles, systematic review, and meta-analysis, in vivo, and in vitro studies were excluded.

The main outcome was nursing empowerment interventions developed for ICU discharge in ICU survivors. The secondary outcomes were the effects of the nursing interventions in this process.

### 2.3. Search Strategies and Data Resources

We reviewed four databases, including Embase, PubMed/MEDLINE, CINAHL, Cochrane Central Register of Controlled Trials (CENTRAL), CUIDEN Plus, and LILACS. The search was conducted on 17 May 2021 and was completed by selecting additional publications from the reference sections of the articles included using the search terms (Table 1). The terms selected were combined using Boolean logical operators (or, and, not). All references were analyzed using Rayyan software (http://rayyan.qcri.org accessed on 10 July 2021) a web-based tool [33]. To ensure thoroughness, we subsequently performed a cited-reference search (“reverse search”) for each article using Google Scholar and reviewed all results of each search.

### 2.4. Review and Study Selection

The review was performed independently by two investigators (CC and RTC) with experience in literature reviews. Primarily, it consisted of reviewing the titles and abstracts of all references retrieved by the database searches (CC and RTC). We searched all articles deemed potentially eligible by one or both reviewers. Secondly, the retrieved full texts were evaluated, and a decision on inclusion or exclusion was made according to the predefined selection criteria (CC and RTC). Any disagreements were resolved by a consensus, and a third reviewer was not necessary. Studies that did not fulfill the predefined criteria were excluded, and their bibliographic details were listed with the specific reason for exclusion.

### 2.5. Data Extraction and Data Synthesis

Two authors (CC and RTC) extracted the data independently and used standardized protocol and reporting forms in duplicate. The following information was extracted from each study included: design, population characteristics, nursing intervention, and results. If relevant data were not included in the article, the researchers searched for more information in the Appendix A, or the author was contacted to request the information.

Narrative methods of synthesis were used to synthesize the included studies. The outcomes were not sufficiently similar enough to perform a meta-analysis. Each study was summarized and described with regard to participants’, characteristics of interventions, the instrument used, and critical outcome results, and this was checked by another reviewer (RTC). One table was created (Table 2).

### 2.6. Methodological Quality Assessment

The risk-of-bias of the studies included was assessed independently using the Cochrane risk-of-bias tool [34]. To minimize bias, studies were graded independently by two reviewers (RTC and YT) and discrepancies were resolved by a consensus, and a third reviewer was not necessary.

## 3. Results

### 3.1. Study Selection

The flow chart of the study selection process is shown in Figure 1. The initial search found that 274 references and 259 studies remained after removing duplicates. After abstract and title screening, 238 studies were excluded. Twenty-one full texts remained, which were assessed for eligibility, leading to the exclusion of 13 studies due to wrong study design (*n* = 7), wrong population (*n* = 4), and wrong outcome (*n* = 2). Eight articles were finally included [35,36,37,38,39,40,41,42]. Therefore, the intervention of the third reviewer was not necessary. The reverse search did not return any additional references.

### 3.2. Characteristics of the Included Studies

Four studies were conducted in England [35,37,39,42], two in the USA [36,41], one in Iran [38], and one in Turkey [40]. Individual study characteristics, including the study design, sample and setting, interventions, and outcomes, are summarized in Table 2.

### 3.3. Participants

In total, 2408 patients were enrolled in the included studies. The sample size ranged from 36 [37] to 1458 [39], and 1108 (46%) of patients received the nursing intervention. The patients’ age included in the studies ranged between 54.6 ± 7.9 [38] and 60.4 ± 15.0 [39] in the nursing intervention group. A summary of patient characteristics is shown in Table 2.

### 3.4. Risk-of-Bias Assessment

There were wide variations in bias in all articles included. The randomization method and allocation concealment were adequate in most trials. In one of the eight trials, participants were blinded to the treatment allocation [37], and in two studies, there was a blinded outcome assessor [35,42]. Most trials, however, lacked blinding of participants, personnel, or outcome assessment (Figure 2). However, close to half of the authors provided insufficient information to assess whether a critical risk of bias existed for other sources of bias. Therefore, the intervention of the third reviewer was not necessary. The results of the quality assessment are shown in Figure 3.

### 3.5. Main Findings

#### 3.5.1. Primary Outcome

Various nursing interventions were made in the studies selected:Information/education interventions. Three RCTs included information skills training programs or educational programs [35,37,40] for patients and family and one “magic empowerment program” with information and education [38] only for patients.Discharge planning. One study assessed discharge needs and defined early discharge planning [36].ICU recovery/therapeutic environment and complex interventions. Three studies included a change in routine by the nurse-led preventive psychological intervention for critically ill patients [39], with hospital rehabilitation, comprising enhanced physiotherapy, nutritional care and information provision, case-management by a ward-based clinical team [42], and an interdisciplinary recovery program with a nurse practitioner and case manager [41].

#### 3.5.2. Secondary Outcomes

Anxiety and depression: Knowles et al. (2009) [37] found that a diary with daily information about patient health reduced anxiety and depression in the experimental group. In the same line, Kuchi et al. (2020) demonstrated changes in patient attitudes toward risk-motivated behavior, and they improved physical health with information and education. Demircelik et al. (2015) found less anxiety and depression in patients receiving nursing education through multimedia interventionPost-traumatic stress disorder: Wade et al. (2019) [39] performed an RCT in 1458 adults post ICU and found no significant differences in PTSD symptom severity at six months among groups.Perceived risk score: Kuchi et al., (2020) [38] in 84 cardiovascular patients, found significant differences between the intervention and control group in the total score of perceived risk and its subscales.Patient satisfaction: Ramsay et al. (2016) [42] assessed the patient satisfaction with the PEQ and revealed significant differences between groups suggesting greater patient satisfaction in the EG.Hospital readmission: Bloom et al. (2019) [41] found that after discharge, at seven days, the readmission rate was 3.6% and 11.6%, in the intervention and control group, respectively. At 30 days, the readmission rate was 14.4% vs. 21.5% in the intervention and control groups, respectively.

## 4. Discussion

We aimed to study nursing interventions based on patient empowerment during ICU discharge and analyze their effects. To the best of our knowledge, this study is the first to systematically review empowerment interventions in patients during ICU discharge.

Few studies to date have analyzed the impact of information and education on patients during their ICU stay and discharge, and most have limitations in the design, sample, and lack of randomization [43]. Patient empowerment studies in other fields have shown that nursing interventions improve patient stress, anxiety, and depression [44,45] (Figure 4). Patients’ emotional states should be evaluated to determine where, how, and when to intervene and ensure that the patient is emotionally prepared for the change between the ICU and the general ward. Situational control is one of the main goals of patient empowerment in this stage [21,23,46]. Patients admitted to the ICU usually feel that they have lost control of their lives, especially those with severe conditions who require sedation and mechanical ventilation, making them totally dependent and unable to decide. Faced with this situation, patients have to adapt to being dependent on others and accept how they carry out procedures, which results in a loss of control of the situation and feelings of helplessness. In addition, complications and slow recovery are, in turn, the cause of delays in transfer to the general ward, increasing daily the feeling of lack of control of the situation. Meleis et al. (2000) found that preparation and knowledge make it easier to empower people for their transition, while a lack of preparation acts as an inhibitor [47]. Therefore, it is necessary to create an environment in which returning control of the situation to the patient is prioritized and in which nurses are responsible for ensuring that patients can receive knowledge according to their expectations [48,49].

In four out of eight studies, the main intervention was information and education for patients and relatives [37,38,40,50]. Various studies have demonstrated the importance of these issues [7,43] and have shown that, when they are lacking, it is more difficult for patients to participate actively during the transition [51].

Four of the evaluated studies explored the impact over patients’ emotional well-being [35,37,38,40]. Knowles et al. (2009) found that a diary with daily information about patient health was beneficial. However, Bench and Day (2015) did not find that written and verbal information during discharge improved patients’ emotional state, specifically anxiety and depression. This may be due, at least partly, to the late intervention; better results might have been achieved if it had been administered early, which could have helped patients to have a more informed perspective.

Other interventions in the review described the determination of patient needs through questionnaires, followed by development of an individualized recovery, and a discharge plan of care with nurse interventions specifically aimed at these needs. Constant evaluation of the needs of patients admitted to the ICU during their stay and transfer to the general ward is necessary to generate an updated and structured care plan that helps ensure the continuity of care, even though drawing up these plans takes time. Kleinpell et al. (2004) demonstrated that such an intervention was associated with patients being better prepared for both ICU and hospital discharge.

Complex interventions addressed towards patient recovery have demonstrated greater patient satisfaction [42] and reduced rates of ICU readmission and mortality [41]. However, Wade et al. (2019) found that nurse-led interventions were not associated with a decrease in PTSD after ICU discharge [39]. Therefore, it is necessary to improve the evaluation and measurement of the effectiveness of these nursing interventions to determine their real benefits and how they could contribute to positive results during ICU discharge.

The evaluation and follow-up of patients during ICU discharge by an advanced practice nurse in a multidisciplinary team was another of the interventions studied. This role appeared in three studies, with two different denominations, including nurse-led [39] and case management [41,42]. Although advanced practice nurses were introduced more than two decades ago as part of the multidisciplinary teams to care for patients with complex needs, they have only been involved in ICU discharge in the last few years. These new roles represent an opportunity to help patients and families regain their sense of control and to cope with the new situation outside the ICU environment.

In the studies included in this review, the term patient empowerment was not explicitly used, but concepts related to empowerment were studied. This result was also found in another systematic review related to empowerment in online communities [50] where 30% of the studies did not use the term empowerment for the intervention. This may be because, despite the various existing definitions of empowerment, the elements that intervene in the concept of power/empowerment and that include control, psychological coping, legitimacy, support, knowledge, and participation, as well as highlight the need for patient empowerment researchers to broaden their perspectives from individual to structural aspects of power and empowerment [51]. In another review of the empowerment concept, the authors proposed that improving the patient’s empowerment would be necessary {Formatting Citation}. In this sense, to change the conceptual and operational ways of considering patient empowerment only as individual and interpersonal elements, but rather patients need a high level of self-efficacy and control of the health situation, and we must integrate the concept of autonomy and the perceived capacity of the patient [52].

Finally, the main contribution of this systematic review is the proposal of nursing interventions to apply the empowerment of the patient during the transition from the ICU. Future studies with designs using rigorous methodologies will increase the quality and credibility of these interventions. Therefore, it is necessary to improve the evaluation and measurement of the effectiveness of these nursing interventions to determine their real benefits and how they could contribute to positive results during ICU discharge.

### 4.1. Applicability of the Findings to the Review Question

Most interventions in this review were carried out by nurses to help ICU survivors. Nurses, when appropriately informed and educated, can apply empowerment interventions to improve the transition from the ICU to the general ward. Our results show the impact on patient empowerment during ICU discharge, with important clinical implications. It is important to detect psychological adverse effects in ICU discharge patients. We suggest that a routine evaluation of anxiety and depression in ICU patients at discharge should be mandatory, as it will permit to carry out a specific intervention to whom they will benefit and assess how beneficial it would be.

However, the concept of empowerment introduced into health should be assimilated and understood by ICU nurses to use it as such and intervene in the patient. In addition, patient dependence on care and needs are not conditions those nurses must automatically and equally assume for all patients transitioning from the ICU to the general ward. Each situation should be evaluated, and each patient’s responses and expectations should be considered individually to ensure adequate care for their needs, taking decision-making and preferences into account. Likewise, the care provided by nurses on the general ward would be much easier if patients had more control of the situation and if they were informed of the changes between the ICU and the general ward [48].

### 4.2. Strengths of the Review

Our systematic review has some strength. We conducted a comprehensive search of the literature, including full-text publications, without language restrictions or filters in the search strategy. Although we included studies published between 2000 and May 2021, it is unlikely that previous relevant trials were missed. The process of the systematic review was rigorous, and all the reviewing authors were appropriately trained and have experience in reviewing manuscripts.

### 4.3. Limitations of the Review

This study has some limitations. The literature review included only articles that had the words “patient” and “empowerment” and “ICU discharge” in the title or abstract and may therefore have excluded some reported interventions on patient empowerment.

## 5. Conclusions

Various nursing interventions during ICU discharge that focused on empowerment were carried out in the studies selected, and this included information, determination of the discharge needs and outcomes of critically ill patients, nursing care plans and assessment, and follow-up by advanced practice nurse. In almost all the studies analyzed, the main intervention was information and education of patients and families. Most of them were associated with benefits from the perspective of controlling the situation and improving negative emotional effects.

### Practice Implications

Nursing interventions using patient empowerment may have positive effects during ICU discharge. This review may help other projects in a similar context to implement new nursing interventions to empower the patient during ICU discharge. In particular, it is important to identify the nursing intervention to contribute to patient empowerment in critical care and, especially, to assess its impact on the different patient dimensions and outcomes.

Future research should focus on the most effective methods of information, education, and patient empowerment during ICU discharge. It would also be useful to conduct more research on interventions that aim to reduce negative effects following transfer, such as structured teaching and information programs. Further research on what the transfer experience means to critical care patients and what effects it has in the immediate post-transfer period is also required. A combination of qualitative and quantitative measures may be needed to evaluate the effect of nursing interventions on patient outcomes.

## Figures and Tables

**Figure 1 ijerph-18-11049-f001:**
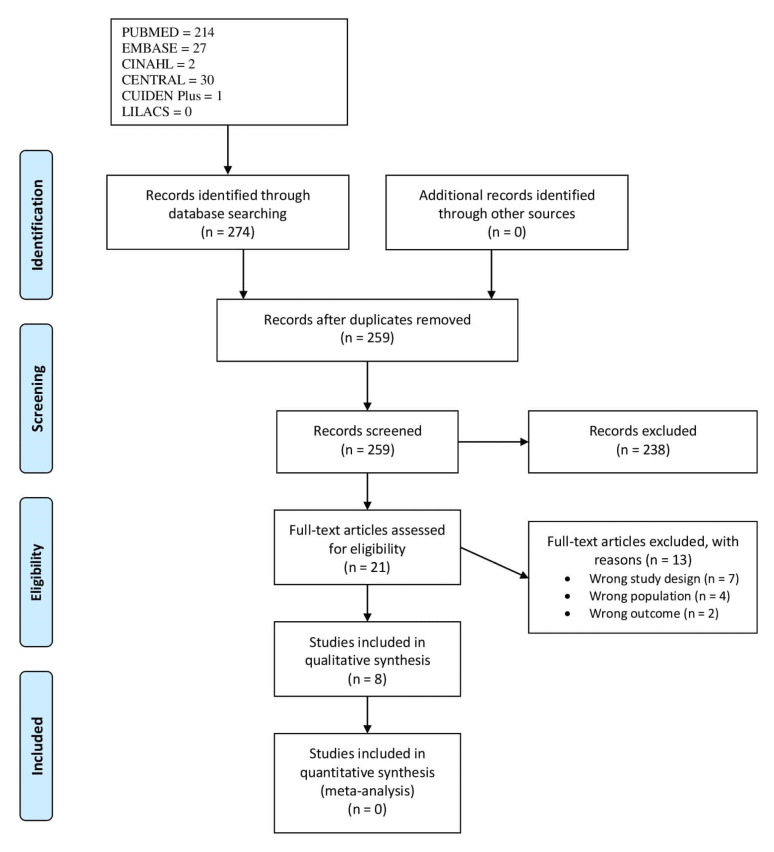
PRISMA Flowchart.

**Figure 2 ijerph-18-11049-f002:**
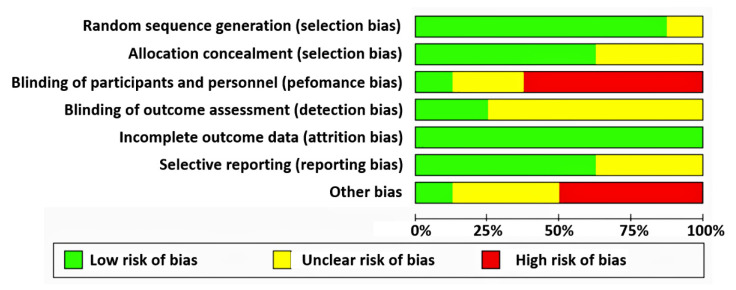
Risk of bias assessment summary.

**Figure 3 ijerph-18-11049-f003:**
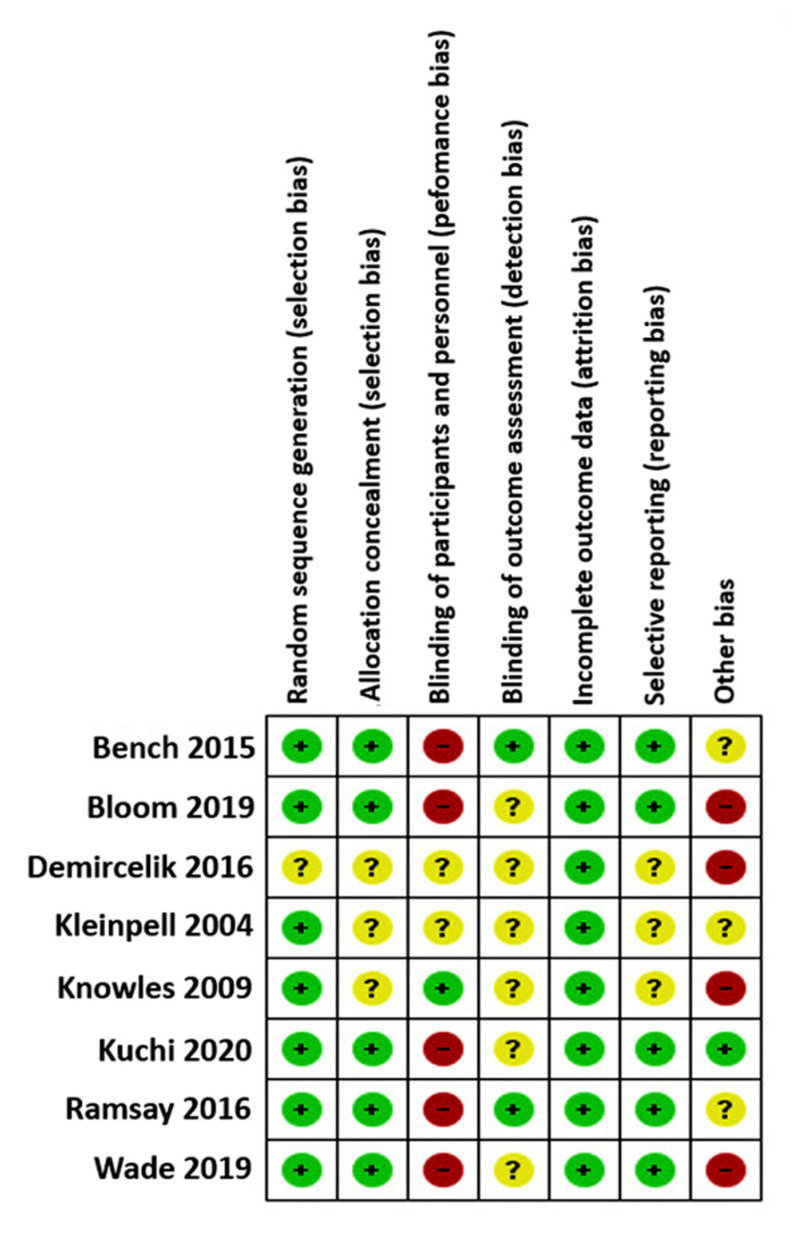
Quality assessment of studies included.

**Figure 4 ijerph-18-11049-f004:**
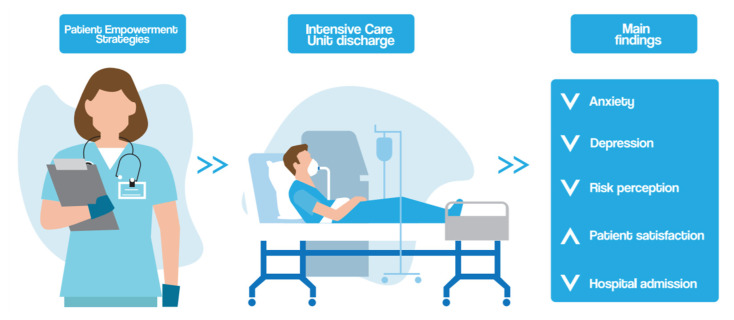
Main findings of patient empowerment strategies performed by nurses.

**Table 1 ijerph-18-11049-t001:** Terms of searching in the different databases and results obtained.

Database and Keywords Combinations	Articles
PUBMED	
((empowerment patient OR patient education OR patient information) AND (ICU discharge OR ICU transfer OR ICU transition) AND (nursing interventions) AND (adults))	214
EMBASE	
((empowerment AND patient OR patient) AND (education OR patient) AND information AND (”ICU discharge” OR (ICU AND (“discharge”/exp OR discharge)) OR “ICU transfer” OR (ICU AND (“transfer”/exp OR transfer)) OR ”ICU transition” OR (ICU AND (”transition”/exp OR transition))) AND (”nursing interventions” OR ((”nursing”/exp OR nursing) AND (”interventions”/exp OR interventions))) AND (”adults”/exp OR adults)	27
CINAHL	
((empowerment patient OR patient education OR patient information) AND (ICU discharge OR ICU transfer OR ICU transition) AND (nursing interventions) AND (adults))	2
Cochrane Library	
((empowerment patient OR patient education OR patient information) AND (ICU discharge OR ICU transfer OR ICU transition) AND (nursing interventions) AND (adults))	30
CUIDEN Plus	
((empowerment patient OR patient education OR patient information) AND (ICU discharge OR ICU transfer OR ICU transition) AND (nursing interventions) AND (adults))	1
LILACS	
((empowerment patient OR patient education OR patient information) AND (ICU discharge OR ICU transfer OR ICU transition) AND (nursing interventions) AND (adults))	0
TOTAL	274

**Table 2 ijerph-18-11049-t002:** Main characteristics of the studies included.

Author, Year, Country	Population	Groups (*n*)	Intervention	Instrument *	Findings/Results
Bench 2015 UK	158 patients (aged >18 years) in two ICUs in England.	CG 1 (59) CG 2 (48) EG (51)	There were three groups; EG received UCCDIP, comprising two booklets, one for the patient with a personalized discharge summary, and one for the family, given prior to discharge to the ward. CG 1 received the usual information and CG 2 received the booklet produced by the ICU steps.	HADS BCOPE PEI	There were no significant differences in psychological well-being measured using HADS, assessed at 5 ± 1 days post unit discharge and at 28 days/hospital discharge among the 3 groups. There were no differences in the other scales after intervention.
Kleinpell 2004 USA	100 patients (aged 65–95 years) in two ICUs of two Midwestern University medical centers.	CG (53) EG (47)	A DPQ was performed to assess discharge needs and define an early discharge planning nurse intervention with formal structured communication to the discharge planning nurse when the patient was transferred from the ICU.	Discharge Adequacy Rating Form SF-36	Patients in the EG were more ready for discharge, more likely to report they had adequate information, and less concerned about managing their care at home than patients in the CG. They also better understood their medicines and danger signals indicating potential complications.
Knowles 2009 UK	36 patients (aged 18–85 years) discharged to medical/surgical wards at Royal Bolton Hospital, Lancashire.	CG (18) EG (18)	ICU diary, containing daily information about their physical condition, procedures and treatments, events occurring on the unit, and significant events from outside the unit.	HADS	Patients in the EG displayed significant decreases in both anxiety and depression compared to CG.
Kuchi 2020 Iran	84 patients (18–65 years of age) with coronary artery disease admitted to post-CCU wards in Tehran hospital.	CG (42) EG (42)	An information and education-based empowerment program following five stages: 1. Motivating patient self-awareness. 2. Assessing causes of problems. 3. Setting goals. 4. Developing personal self-care plans. 5. Assessing achievement of goals.	SAQ Perception of Risk of Heart Disease Scale.	There were significant differences between the two groups in total score of perceived risk and its subscales. The intervention changed patients’ attitudes toward risk-motivating behavior change and improving physical health.
Wade 2019 UK	1458 patients (>18 years of age) in 24 general ICUs in UK	CG (789) EG (669)	Nurse-led preventive psychological intervention for critically ill patients, comprising three phases: 1. Creating a therapeutic environment in ICU. 2. Three stress support sessions for patients screened as acutely stressed. 3. Relaxation and recovery program for patients screened as acutely stressed.	PTSD Symptom Scale–Self-Report. STAI-6. HrQoL	There were no significant differences in PTSD symptom severity at 6 months among groups. There were no differences in the other scales after intervention.
Ramsay 2016 UK	240 patients (>18 years of age) in Edinburgh, Scotland	CG (120) EG (120)	A complex intervention aimed towards post-ICU rehabilitation delivered between ICU and hospital discharge by dedicated rehabilitation assistants (RAs) working together with existing ward-based clinical teams. The intervention comprised: enhanced physiotherapy nutritional care information provision case-management	PEQ HRQoL SF-12 HADS Davidson’s Trauma Scale	The PEQ revealed significant differences between groups, suggesting greater patient satisfaction in the EG. Focus group data strongly supported and helped to explain these findings. There were no differences in the other scales after intervention.
Demircelik 2015 Turkey	100 patients, Turkish coronary ICU	CG (50) EG (50)	Multimedia nursing educational intervention.	HADS	There were significantly higher decreases in HADS scores in the EG.
Bloom 2019 USA	232 patients (≥18 years of age) at Vanderbilt University Hospital.	CG (121) EG (111)	Interdisciplinary ICU recovery program, comprising: inpatient visit by a nurse practitioner an informational pamphlet a 24/7 phone number for the recovery team an outpatient ICU recovery clinic visit with a critical care physician, nurse practitioner, pharmacist, psychologist, and case manager	Death and readmission rate	Hospital readmission after discharge at 7 days (3.6% vs. 11.6%) and 30 days (14.4% vs. 21.5%), median time to readmission (21.5 (IQR 11.5–26.2) vs. 7 (4–21.2) days) and the composite outcome of death or readmission within 30 days of hospital discharge (18% vs. 29.8%) were significantly better in the ICU recovery program group than in the usual care group.

* Instrument used to assess the impact of the intervention. Abbreviations: BCOPE: Brief Coping Orientations to Problems Experienced, CG: Control group, DPQ: Discharge Planning Questionnaire, EG: Experimental group, HADS: Hospital Anxiety and Depression Score, HrQoL: Health-related Quality of Life, ICU: Intensive Critical Unit, ICU steps: Intensive care guide for patients and relatives, IQR: Interquartile range, SAQ: Seattle Angina Questionnaire. PEI: Patient Enablement Instrument; PEQ: patient experience questionnaire, PTSD: Post Traumatic Stress Disorder, SF-12: Short Form 12 Health Survey, STAI-6: State Trait Anxiety Inventory 6-item version, UCCDIP: User-Centered Critical Care Discharge Information Program, UK: United Kingdom.

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
