# Peer review of "Nursing Interventions for Patient Empowerment during Intensive Care Unit Discharge: A Systematic Review"

_ijerph, 2021, doi:10.3390/ijerph182111049_

Round 1
Reviewer 1 Report
Introduction: A paragraph should be presented to highlight the nurse's role in patient discharge. The final paragraph of the introduction mixes, study design, objective and research question. If you want to formulate the objective as a question, do so in PICO format.
Methods. The methodology does not exhaustively follow the PRISMA methodology. Add the type of design that was in the introduction.
In the section: Criteria for inclusion in the review no exclusion criteria are presented. However, in the section Review and study selection the same is presented. Properly following the structure of the review would be appropriate.
In the section: Databases, and being the authors from Spain, it is surprising that they have left out the two main Spanish and Portuguese language health sciences databases: LILACS and CUIDEN Plus, which may represent a serious selection bias. Can the authors justify the reason for this? It would be important to rule out this document selection bias. In fact, 6 of the 8 studies included are from the English-speaking world. Are there no studies in other contexts or have they not been searched for?
Another key element to accept the publication and determine if there were more biases is in the review and study selection section. The authors should specify more clearly how they decided whether or not to include the studies. In addition to complying with the inclusion criteria (and not having exclusion criteria)
They state that the Cochrane Review Tools were used to evaluate the methodological quality of the studies. But there is no record of whether the intervention of the third reviewer was necessary in many articles.
Was a reverse search performed with the studies found?
Results: If the exclusion criteria are clear in the flow diagram, it should explain which of the 238 studies eliminated were for each criterion.
Discussion: Sometimes there is a reiteration with the results. Review.
Author Response
the response is attached

Reviewer 2 Report
Please mention the review aspects using PICO in the methods section.
There is a need to the precise description of inclusion and exclusion criteria.
Did you include grey literture?
You performed the risk of bias assessment. How about the quality assessment process? Did you use instruments to assess the quality of the articles?
How did you synthesised the research results? The process of data analysis and research synthesis should be decsribed in the methods. It would help to make Figure 1 sense to readers.
Why a meta-analysis could not be conducted?
The quality assessment result should be described with detail in the result section.
A figure can help to summarise your main review findings.
You are suggested to compare your findings with similar systematic review in the discussion rather than simple qualitative studies or clinical trials.
Author Response
The response is attached

Round 2
Reviewer 1 Report
These considerations were made in the first review.
In the section: Databases, and being the authors from Spain, it is surprising that they have left out the two main Spanish and Portuguese language health sciences databases: LILACS and CUIDEN Plus, which may represent a serious selection bias. Can the authors justify the reason for this? It would be important to rule out this document selection bias. In fact, 6 of the 8 studies included are from the English-speaking world. Are there no studies in other contexts or have they not been searched for?
They state that the Cochrane Review Tools were used to evaluate the methodological quality of the studies. But there is no record of whether the intervention of the third reviewer was necessary in many articles.
Was a reverse search performed with the studies found?
The authors have not justified neither in the document nor in the letter the reason for them.
I believe they are necessary for acceptance
Reviewer 2 Report
The quality of figures in the article is not good. They should be replaced with high quality ones.
